# PUT-Hand—Hybrid Industrial and Biomimetic Gripper for Elastic Object Manipulation

**Tomasz Mańkowski** *, **Jakub Tomczyński**, **Krzysztof Walas and Dominik Belter**

Institute of Robotics and Machine Intelligence, Poznań University of Technology, ul. Piotrowo 3A, 60-965 Poznań, Poland; jakub.tomczynski@put.poznan.pl (J.T.); krzysztof.walas@put.poznan.pl (K.W.); dominik.belter@put.poznan.pl (D.B.)
* Correspondence: tomasz.mankowski@put.poznan.pl

**Abstract:** In this article, the design of a five-fingered anthropomorphic gripper is presented specifically designed for the manipulation of elastic objects. The manipulator features a hybrid design, being equipped with three fully actuated fingers for precise manipulation, and two underactuated, tendon-driven digits for secure power grasping. For ease of reproducibility, the design uses as many off-the-shelf and 3D-printed components as possible. The on-board controller circuit and firmware are also presented. The design includes resistive position and angle sensors in each joint, resulting in full joint observability. The controller has a position-based controller integrated, along with USB communication protocol, enabling gripper state reporting and direct motor control from a PC. A high-level driver operating as a Robot Operating System node is also provided. All drives and circuitry of the PUT-Hand are integrated within the hand itself. The sensory system of the hand includes tri-axial optical force sensors placed on fully actuated fingers' fingertips for reaction force measurement. A set of experiments is provided to present the motion and perception capabilities of the gripper. All design files and source codes are available online under CC BY-NC 4.0 and MIT licenses.

**Keywords:** robotic hand; control; perception; tactile sensing; mechatronics; grasping; manipulation; PUT-Hand; underactuated

## 1. Introduction

Manipulation of elastic pipes and wires in factory environments is performed mainly by human operators. Some of the tasks on the elastic objects are performed by the machines designed specifically for that operation. Rarely, the manipulation of elastic objects is performed by robots which can adapt to the changes in the process. This scenario is still challenging, due to the deficiencies in mechanical design of the grippers and the perception systems. Interaction with elastic objects requires not only high manoeuvrability but also reliable tactile feedback.

Robots have a potential to manipulate elastic objects autonomously without a supervision of a human operator. Application of robots in factories improves the production process and reduces number of errors and mistakes made by humans. However, manoeuvrability of the robot and capability to work with elastic objects still need development.

Capabilities of the grippers available in the industry are limited, as most of them are two-fingered or three-fingered [1–3]. They are often fully actuated systems with position-based control, designed for precise manipulation. This means that they are not designed to deal with variability and uncertainty of an environment, a slight change in object positioning or its shape may cause a failure.

Contrarily, many adaptive grippers are designed to replace missing body parts, mimicking the human hand, with five fingers [4]. Digits can be both fully actuated [5] or underactuated for better

adaptation to shape of grasped objects [6]. However, they are designed to replace human hands, not to operate as a gripper of an industrial robot.

In this research, we focus on the design of a robotic hand which can be used to manipulate elastic objects. We carefully study the literature to find a compromise between precision of the grasp and adaptability, achieved by optimising the number of fully actuated and underactuated fingers. We also study the perception system of robotic hands and propose the application of optical tactile sensors on the fingertips. We present the application of the hand in several tasks related to manipulation of elastic objects, including grasping, tactile sensing, and application-oriented tasks.

### 1.1. Problem Statement

Our main goal was to design a compact robotic hand which has the capability to interact with elastic objects. The hand should enable precise manipulation and grasping, while allowing power grasping with underactuated fingers. We had to determine which joints of the fingers should be fully actuated and which ones should be underactuated. Moreover, we had to design the actuation mechanisms to preserve compact dimensions of the hand.

The design of the hand should provide full observability of all joints, including underactuated ones, as the state of the joint does not depend only on the drive, but also on the shape of grasped object, and the state of neighbouring joints. The perception system of the hand should also allow measuring mechanical properties of objects and reaction forces.

Finally, the mechanical design and perception system should allow performing task-specific modelling. During interacting with elastic objects the sensors should provide information about the contact with the object and measure the changes in the object's state. The example scenario is plug insertion, during which reaction forces increase until the plug reaches a stable configuration. The final state of the system should be detected by the sensory system of the proposed hand.

### 1.2. Approach and Contribution

In this article, we present a new open-source design of the dexterous robotic hand - PUT-Hand, shown in Figure 1. The gripper includes fully actuated fingers with tactile sensing, and underactuated fingers for power and adaptive grasps. The mechanical part is providing in one design the compact human-like hand for skilful manipulation with hybrid fully actuated and underactuated fingers. The hand is lightweight and has a unique feature that all the drives are built-in the palm of the hand, which is reducing the length of the end effector hence it is minimally shrinking work volume of manipulator equipped with our hand. The hand is reproducible. The mechanical and PCB design files, firmware, and ROS node are available at https://github.com/puthand under CC BY-NC 4.0 and MIT licenses. We also provide a broad literature review related to robotic and prosthetic hand designs. We also contribute to the control and perception system of the hand, presenting hand on-board controller and a ROS driver. The ROS driver enables computing forward and inverse kinematics, motion planning, and visualisation. The full software stack is available.

The hand is designed for elastic object manipulation by providing three fingered-like manipulation when thumb in the first joint is oriented towards the inner part of the palm. The hand operates in robotic gripper mode and it allows in hand dexterous manipulation of elastic elongated objects i.e., cables and hoses. Additionally, thanks to full sensorisation of the hand, joint encoders and force sensors at the fingertips, one can estimate elastic object parameters by performing in hand manipulation and measuring fingers displacement and reaction forces caused by the movement of the fingers.

Finally, we provide an experimental verification of PUT-Hand and its controllers. We show manoeuvrability of PUT-Hand during grasping various objects. We demonstrate that the sensory system of the hand can be used to detect contact with an object but also to measure its physical properties (e.g., object stiffness). We also provide application-oriented experiments to show the interaction of the hand with elastic objects during pipe bending and inserting a plug into a socket.

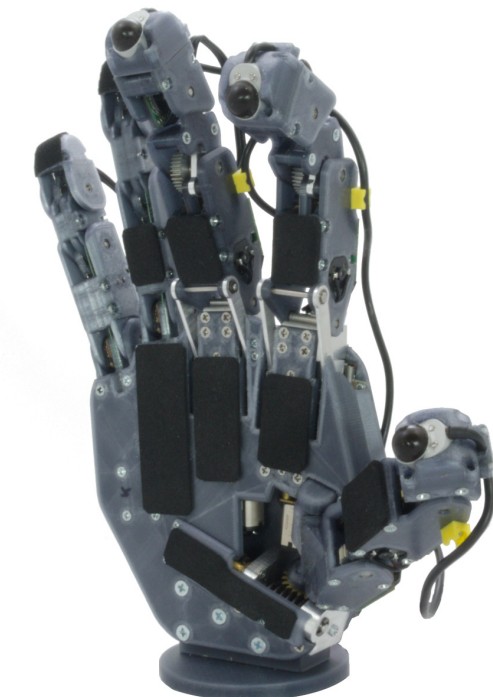

**Figure 1.** PUT-Hand—An open source dexterous robotic hand.

## 2. Related Work

We limit the review to hand designs available in the literature, also considering commercially available hands only if description and specification are detailed enough. For this reason, we do not include some of the commercial prosthetic hands, such as VINCENT hands by Vincent Systems [4], iLimb and iLimb Pulse by Touch Bionics [7], and Michelangelo by Ottobock [8]. If a research team publishes a series of designs, we only consider the latest version in the review. Multiple hand designs by the same group are only considered if the mechanical concept of the hands differs significantly.

Most of the hands considered in the review are research platforms presenting unique mechanical, control or sensing concepts. Additionally, we include five commercial platforms in the comparison. In the review, we have proposed gripper design taxonomy based on three criteria: joint drive mechanism type, whether the system is fully actuated or not, and number of digits and kinematic structure.

### 2.1. Joint Drive Mechanisms

A large variety of drive to joint power transfer methods can be found in the literature. In this work we analyse the most common and representative approaches. The classification of joint drive mechanisms for robotic fingers is presented in Figure 2 as eight categories: direct drive with gears, fully actuated joints with tendons, backdrivable tendon, underactuated finger with a tendon, rigid linkage system, underactuated joints with rods, pneumatically actuated, and pneumatic/hydraulic muscles. We identified 21 structures with tendon mechanisms [2,3,6,9–26], two with direct drive based on gears [1,5], one rod-based [27], two with rigid linkage system [28,29], one with differential drivetrain and cams [17], one with pneumatic chamber [30], and one with hydraulic muscles [31]. All the cited papers were grouped in Table 1.

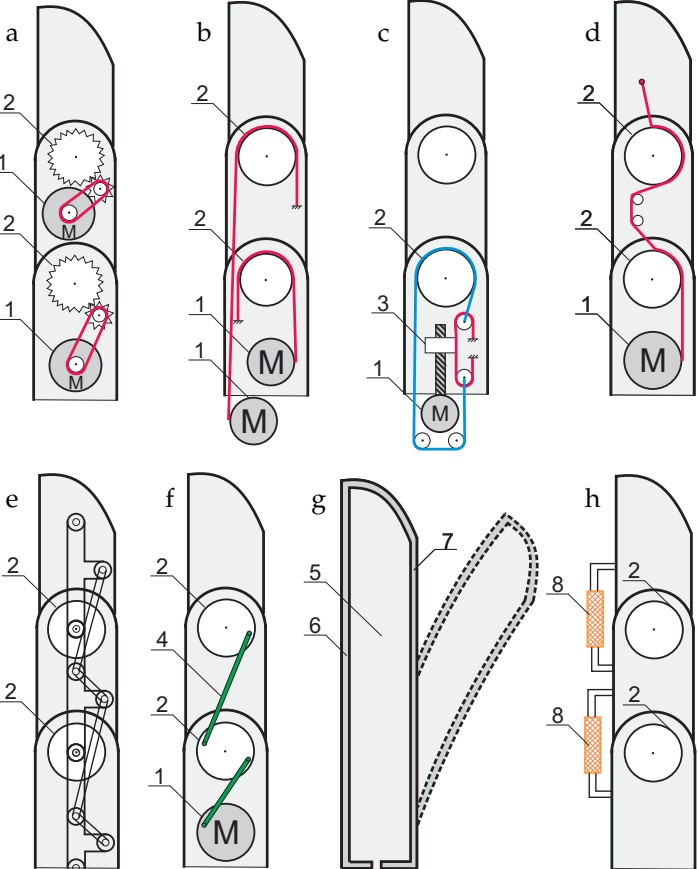

**Figure 2.** Joint drive mechanisms for robotic fingers: direct drive with gears (**a**), fully actuated joints with tendons (**b**), backdrivable tendon (**c**), underactuated finger with tendon (red or blue) (**d**), rigid linkage system (**e**), underactuated joints with rods (green) (**f**), pneumatically actuated (**g**), pneumatic/hydraulic muscles (**h**). Legend: 1—electrical motor, 2—joint, 3—nut, 4—rod, 5—air chamber, 6—elastic fabric, 7—inelastic fabric, 8—pneumatic/hydraulic muscles.

**Table 1.** Related work—joint drive mechanisms review.

| | |
|---|---|
| Tendon mechanism | [2,3,6,9–26] |
| Gears | [1,5] |
| Rod-based | [27] |
| Rigid linkage system | [28,29] |
| Differential drivetrain | [17] |
| Pneumatic chamber | [30] |
| Hydraulic muscles | [31] |

The most popular method for joint coupling uses tendons (Figure 2b). The tendon transfers the torque from an electrical motor or force from a linear actuator to the rotational joint. The tendon might be coupled with a spring to generate compliant behaviour of the fingers. The popularity of tendon-like mechanisms comes from the fact that this mechanism is bio-inspired and gives the natural compliance and adaptation of the hand. This approach reduces the number of actuators and allows obtaining a robust grasp without additional tactile or force/torque sensors.

Gears are used mainly in the fully actuated fingers to transfer the torque from the drive to the joint [9,11,31]. Rods become popular when the drives are located in the palm and the rods push the finger links [27]. This approach reduces the weight of the fingers. The lighter hands are actuated by the pneumatic chambers [30]. In this case the fingers are made of elastic material and bend when the pressure inside the chamber changes. Moreover, the finger adapts to the shape of the object [30]. On the

other hand, the gripper with pneumatic chambers requires the air compressor. The same problem exists when pneumatic [22] and hydraulic [31] muscles are used.

The fingers of PUT-Hand for the first three fingers are fully actuated with rigid linkage system and gears [28,29]. The last two fingers are tendon driven, e.g., [15].

*2.2. Mechanical and Kinematic Structure*

In the review we identified six grippers which are fully actuated [1,9,11,22,31,32]. However, most of the designs exploit the underactuation concept: [2,3,6,14–16,18–21,23–25,27–30,33], ref. [5,10] (two joints coupled), ref. [12] (four joints coupled), ref. [13] (20 joints/9 actuated), ref. [17] (18 joints/16 actuated). Furthermore, the review revealed the following categories with respect to digit count: three-fingered [1–3], four-fingered [6,9,11,25,34]. Most of the grippers described in the literature have five fingers: [5,10,12–17,17–24,27–32]. All the hand structures were grouped in Table 2 regarding mechanical structure and in Table 3 regarding kinematic structure.

**Table 2.** Related work—Robotic hands mechanical structure review.

| | |
|---|---|
| Fully actuated | [1,9,11,22,31,32] |
| Underactuated | [2,3,6,14–16,18–21,23–25,27–30,33] |
| Hybrid designs | [34–41] |

**Table 3.** Related work—Robotic hands kinematic structure review.

| | |
|---|---|
| Three-fingered | [1–3] |
| Four-fingered | [6,9,11,25,34,39,40] |
| Five-fingered | [5,10,12–17,17–24,27–32,35] |

In our design we decided to use five fingered design closely resembling human hand in geometry and size as it was the outlined in [35]. This choice allows us to perform dexterous manipulation and different grasps and manipulation strategies are obtained in software without changes in the mechanical hardware of the hand.

2.2.1. Fully Actuated Designs

Most of commercially available grippers have a limited number of joints, with full actuation and position sensing (e.g., 3-finger Schunk SDH [1]). In a backdrivable CEA hand [10] the joints are fully actuated, with last two joints in each digit mechanically coupled, providing natural behaviour of the hand during force control and high robustness. Utah/M.I.T Dextrous hand [9] is a 4-fingered tendon-driven design, where 32 drives are required. Another 4-fingered design, Sandia hand, was designed by Quigley et al. [11], with each joint actuated by brushless DC servomotors and tendons. Focus was put on the robustness of the hand, and fingers separate from the hand in case of a collision. The UB Hand 3 [17] has 16 degrees of mobility, with joints driven by tendons.

Grasping control algorithm problems become apparent in more complex fully actuated 5-fingered designs [5]. The gripper with the most complex kinematic structure which is considered in this review is Shadow hand [12]. The hand has 5 fingers and 24 joints actuated by McKibben muscles and tendons, which makes it robust to disturbances, but also mechanically complex and of substantial size and weight. The latest hand built at the University of Washington has a similar kinematic structure, with two versions built: driven by pneumatic cylinders [22] and driven by servomotors [23]. Both versions use tendons to transfer the energy from drives to the joints and the drives are mounted in the forearm.

When using a fully actuated design, all joints angles have to be computed according to given dimensions of handled object. If the measurements are uncertain the hand has limited capabilities to adapt to the shape, or the adaptation is slow due to delayed feedback from sensors. On the other hand, fully actuated grippers can be used to perform more complex manipulation tasks. The digits can

reach any position in the workspace in contrast to underactuated systems, where motion trajectory also depends on the shape of the manipulated object.

### 2.2.2. Underactuated Designs

Underactuation concept not only allows designing compliant hands which adapt to the shape of the objects but also to reduce the number of actuators and weight of the hand. One of the first underactuated anthropomorphic grippers was built at the University of California and Belgrade [14]. The hand has 5 fingers and 14 joints, driven by two servomotors and a tendon-driven mechanism. The elastic coupling of the joints allows the fingers to adjust to the shape of an object. TBM hand [28] is a five-finger design using a rigid linkage system to couple joints, all joints are driven by a single DC servomotor. Compliance is obtained by application of extension springs pulling the linkage systems. Similarly, ref. [20] is an underactuated structure driven by one DC motor.

Four fingered design is described in [25] the project uses simple 3D-printed components with compliant flexure joints and off-the shelf parts to provide low-cost, open-source underactuated hand.

Three-fingered designs often mimic the behaviour of the thumb, index, and middle finger of the human hand. An underactuated prosthesis proposed by Zollo et al. [2] is optimised to be capable of reproducing natural human motions. Similarly, the 3-fingered RTR II hand [3] uses differential mechanisms to control multiple joints. The BarretHand designed for industry [42] uses the mechanism which shifts torque to appropriate finger joint. The joints are locked when the fingertip reaches contact with the object. The number of actuators is limited in SRI hand [6] and two fingers can be rotated on a slider mechanism. Breaks in joints are used to reduce power consumption and motor torque during a steady grasp. The same authors also propose the tool that is aimed to design underactuated hands [43]. The simulation tool takes into account the dynamics of the hand, an actuation mechanism, and contact friction. The core of the simulation engine is based on three-dimensional force fields. Southampton-Remedi hand [29] has six drivable degrees of freedom. The authors of [24,44] explore the concept of active synergies in performing grasping and manipulation tasks. The compliance of SmartHand [18,19] is obtained using Hirose's soft (differential) finger mechanism [45]. Various hands were proposed by groups which are interested in anthropomorphic prosthetics. An underactuated MANUS hand [16] has five fingers but only three of them are actuated. The hand has four grasping modes: cylindrical, precision, hook and lateral obtained with two servomotors only. Due to simplicity in mechanical design underactuated grippers can be easily prototyped using 3D printing [27,46].

Underactuated designs have a reduced controllability compared to fully actuated ones; however, the general trend is to reduce the number of actuators. The Vanderbilt hand [21] has five fingers and 16 joints driven by tendons and five DC motors. The UNB anthropomorphic hand [33] has five fingers and only three DC motors. The complex motions are obtained using differential drivetrain and cams.

To achieve required dexterity while keeping the design compact and control algorithms simple, hybrid designs combining both approaches can be explored, with fully actuated fingers dedicated to precise manipulation and underactuated supporting fingers for power grasp. One of the possible solutions was presented in [34], where authors presented a unique design of the fingers which allows for generating linked and adjustable motions. Joints exhibit a coupled movement in free space and moves adaptively when in contact with the objects. Word hybrid applied to hand-design is understood differently in the literature. For Mizushima et al. hybrid was used in the context of hybrid design of the fingers, which skeleton was tendon driven and inside the skeleton the granular material after removing the air the grasping posture can be fixed [36]. A similar understanding of word hybrid is present in [38]. In the case of work presented, in [37] word hybrid is used for describing the hand with three mechanical and three soft fingers. Additionally, in work [35] word hybrid was used to characterise linkage and tendon driven-based fingers. In work done by Jeong and Cheong, the hybrid nature of the hand is understood as the mode of operation. Hand uses four fingers for human-like motions in human hand mode, and three fingers without the thumb when it is used in conventional robotic hand mode [39,40]. Conversely, Cerruti et al. use word hybrid to describe two actuation

system present in hand and working in parallel. One is responsible for gesture capability but low force (linkage mechanism) and the second is used when the grasping force is needed (tendons) [41].

In the case of PUT-Hand, the hybrid is understood two folds. First, the hand is composed of three actuated fingers and two underactuated. Second, PUT-Hand, like the design proposed in [39,40], has two modes of operation, which in case of a PUT-Hand depends on the orientation of the thumb. When the thumb in the first joint is oriented towards the inner part of the palm the hand operates in robotic gripper mode similar to BarretHand [42]. This configuration allows the robot to perform in hand manipulation of the elastic objects such as cables. When the thumb is rotated outwards from the palm, it resembles a human hand as in [35].

### 2.3. Tactile Sensing

A suitable sensory system is crucial while performing successful grasping and manipulation tasks, particularly for elastic object handling. In autonomous, unsupervised manipulation, initial gripper configuration can be chosen based on an object model obtained by a RGB-D sensor [47,48]. Vision-based grasping algorithms can be further improved by active-sensing—a strategy for view selection to maximise the surface reconstruction and safety of the planned trajectory [49]. However, data from depth camera can be noisy, causing uncertainty in generated model [50], leading to a decrease in overall autonomous manipulation performance or even a failure.

Equipping a gripper with tactile sensors can further improve its manipulation capabilities in robotised setups by providing feedback to the robot controller [51], estimating the contact force and actively controlling the reaction forces to stabilise the grasp [52], or as a source of object identification [53,54]. Tactile information can also be used to determine physical properties of an elastic object or to determine the state of manipulated object. Tomo et al. [55] has proposed uSkin 3D tactile sensors intended for Allegro Hand fingertips and phalanges, providing 16 independent force measurements for each contact plane. Data from these sensors can be used as an input for Convolutional Neural Networks in tasks of object identification [56].

Other designs introduce tactile sensing using: inertial units [57], piezoelectric sensors [58,59], resistive sensors [60], a camera for deformation measurement [61], or capacitive sensors [62,63]. An extended literature review in the field of touch sensing is presented in [64,65].

## 3. PUT-Hand Design

### 3.1. Mechanical Design

The main design goal of PUT-Hand project was to create an anthropomorphic gripper which is capable of performing both precise object manipulation and power grasps, in an industrial environment, while using as many off-the-shelf parts as possible. A taxonomy of human grasps [66] shows that most of precision grasps and manipulations is done using three or less fingers—thumb, index, and middle. Many grasps use virtual fingers, where several fingers work as one functional unit. Ring and little fingers are used mostly as assisting fingers in power grasps, where individual control of each joint is not necessary. To provide a balance between grasping capabilities and complexity of the gripper's mechanical and control structures, a hybrid structure was proposed. The design incorporates three fully actuated fingers (thumb, index, and middle), and two underactuated fingers. All movable joints, in both actuated and underactuated fingers, are fitted with position sensors and observable, allowing for full grasping planning and simulation. The resulting design, shown in Figure 3, is mostly 3D-printable, with single elements requiring CNC machining from aluminium or turning from stainless steel. All drives and controllers are integrated within the palm or fingers.

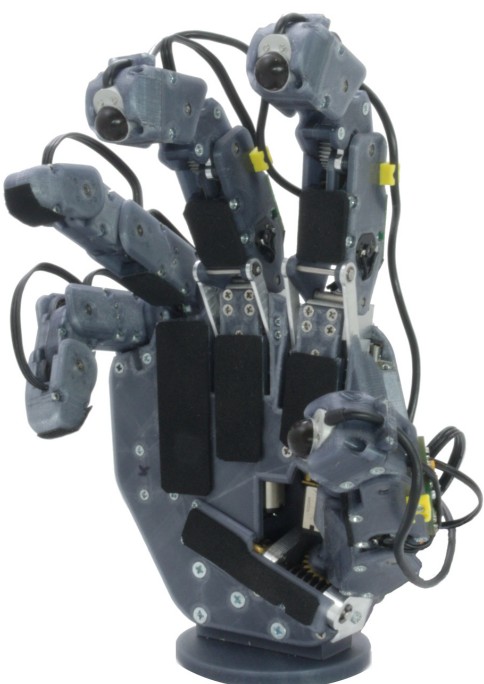

**Figure 3.** PUT-Hand—an open-source dexterous robotic hand.

### 3.1.1. Fully Actuated Fingers (Index and Middle)

Index and middle fingers are designed as fully actuated, with two degrees of freedom. MCP is driven independently, while PIP and DIP share the same drive. This configuration provides full control over fingertip position in finger's 2D plane while maintaining mechanical simplicity (MCP is a single DoF joint with no adduction or abduction ability). The fingers share the same design and dimensions, allowing for easier manufacturing and parts interchangeability. Overall finger dimensions of 18 mm (width) by 20 mm (height) closely correlate with adult mean index finger width [67]. Annotated design of the finger is shown in Figure 4.

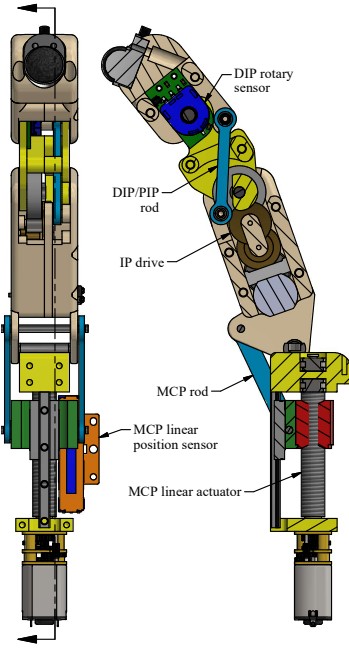

**Figure 4.** Mechanical design and drives of a fully actuated finger; design intended for index and middle fingers.

Due to large torque requirements in MCP, a linear actuator was designed to drive the joint. The drive uses a Pololu Micro High Power 6 V DC motor with 1:75 gearbox driving an Igus DryLin$^{\circledR}$ lead screw (2.54 mm pitch). The lead screw is supported by a set of two thrust bearings on the opposite side. The lead nut is attached to a feed sliding on a CPC MR3ML ball bearing linear guide. Drive position feedback is obtained from an Alps RDC1022A05 linear resistive position sensor.

The linear actuators fit within the palm of the gripper and flex the fingers using aluminium rods. The relation between actuator position and finger flexion is shown in Figure 5. The structure reaches maximum efficiency at $\theta_{MCP} = 61°$, exerting the force of 8.7 N at the fingertip.

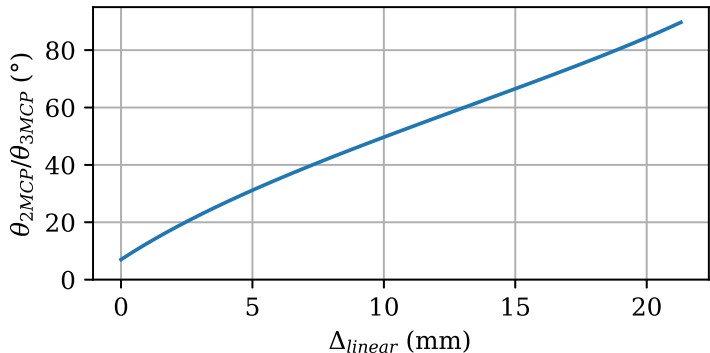

**Figure 5.** Relation between linear actuator movement and finger flexion in MCP.

The second drive resides inside the proximal phalanx. The drive uses motor and metal gears from Hitec (gear set 56,396), used in a range of miniature digital servos (e.g., HS-5245MG). The gearbox was reconfigured to fit in narrow space of the phalanx and drives PIP directly. Movement is passed to DIP using an aluminium rod. Angle feedback is provided at DIP using an Alps RDC503013A rotary resistive sensor. Relation between PIP and DIP angles is shown in Figure 6. The rotation ratio between PIP and DIP was chosen, so with both drives fully flexed, the fingertip touches the metacarpus. When fully extended, the drive exerts the force of 8.0 N at the fingertip.

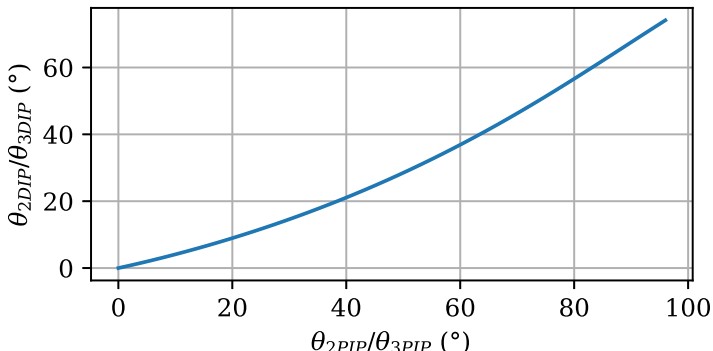

**Figure 6.** Relation between flexion in PIP and DIP.

The finger features an interchangeable fingertip, allowing for mounting of a 3D OptoForce OMD-10 force transducer (shown in the drawings) or a resin moulded passive fingertip, according to particular task requirements.

### 3.1.2. Thumb

Inclusion of an opposing thumb is vital to the performance of many types of grasps. The thumb has three degrees of freedom—two joints in planar configuration (MCP and IP), and third CMC placed at an angle. Thumb's MCP and IP drive share a similar design to index and middle fingers' PIP. The DC motors and gearing are the same, but gearbox layout was altered to fit in the available space in metacarpal and proximal phalanx. Thumb design is shown in Figure 7.

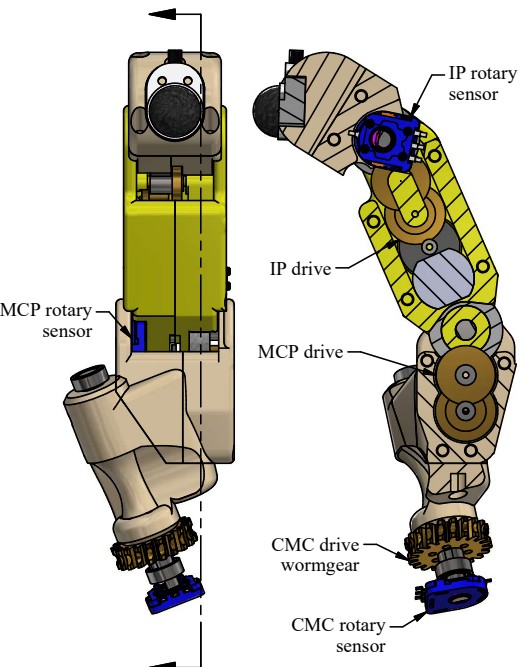

**Figure 7.** Mechanical design and drives of fully actuated thumb; CMC drive motor and worm (not visible) are enclosed in the metacarpus.

Due to the nature of CMC operation, which is moved mainly when switching between various grasping types, a worm gear drive was used. The drive uses a Pololu Micro High Power 6 V DC motor with 1:298 gearbox with an additional 1:20 reduction provided by the worm drive. Resistive rotary sensors by the Alps were used to provide angle feedback from all joints.

3.1.3. Underactuated Finger

Ring and little fingers were designed as underactuated structures to simplify both mechanical design and control algorithms. The fingers use a tendon flexion mechanism, as seen in Figure 2d. The tendon is made from a braided fishing line, providing low extensibility, while maintaining high compliance. Each tendon is wound onto a spool driven by a Pololu Micro High Power 6 V DC motor with a 1:298 gearbox.

An eccentric ring is placed around each joint axle and attached to a pair of helical extension springs. While in other approaches springs have been used to store potential energy for efficiency improvements [68], here they are required to return the finger to its extended position and distribute flexion evenly among the three joints. To limit the number of required off-the-shelf part types, all joints share the same type of springs, with only the anchor radius of the spring differentiating the extension torque. The torque was adjusted experimentally to allow the fingers to straighten in the most demanding orientation (with palm facing downwards), without adding too much additional resistance to the drive. The overall design of the finger is shown in Figure 8. Both fingers share the same design, but proximal and intermediate phalanges are shorter in the little finger.

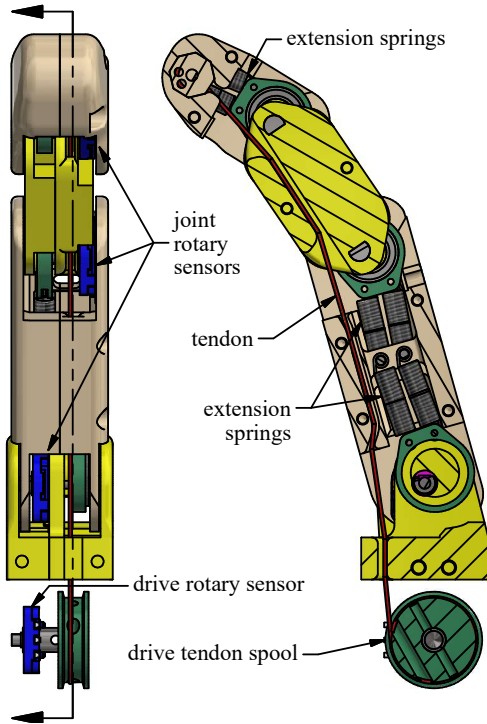

**Figure 8.** Mechanical design of an underactuated finger; design intended for ring and small fingers; drive motor not visible.

To provide full joint state feedback to motion planning software, each joint is equipped with a resistive rotary sensor. An additional rotary sensor is mounted at the spool shaft to enable proper drive control. To summarise the mechanical concept of the PUT-Hand a short comparison with typical robotic hand is provided. Namely, DLR-HIT Hand II [5] and Pisa/IIT Softhand 2 [24]. DLR-HIT Hand II is fully actuated with 15 degrees of freedom, three in each finger. There are five identical fingers. All the drives are embedded in the palm. The motion of each finger is controllable. On the contrary, the PISA IIT Softhand 2 has 19 anthropomorphic degrees of freedom controlled by one motor and relies fully on underactuation concept. PUT-Hand exploits the best of two worlds. It is fully actuated, as DLR-HIT Hand II, for the first three fingers. These fingers are needed for precise manipulation and force sensing. Additionally, it is exploiting underactuation, as Pisa/IIT Softhand 2, for the last two fingers which are supporting the grasping of larger objects.

### 3.2. Controller

The low-level controller of PUT-Hand was designed with high modularity in mind. The main unit, called HUB, is responsible for communication with the high-level controller (e.g., a PC), communication with individual servomechanisms, and (optionally) internal drive position control. Each servomechanism (referred to as DRIVER) is a separate, independent module consisting of a direct-current (DC) motor, an adequate number of encoders, and a printed circuit board with all necessary communication and motor control circuitry. Overall controller architecture is shown in Figure 9. DRIVER modules are connected to the HUB using a bus configured in a star pattern, which allows for easy replacement of faulty drives integrated into the mechanical design of the hand. Controller attachment and arrangement on PUT-Hand is visible in Figure 10. The system is fully scalable and can be used in designs with various drive configurations, not limited to grippers.

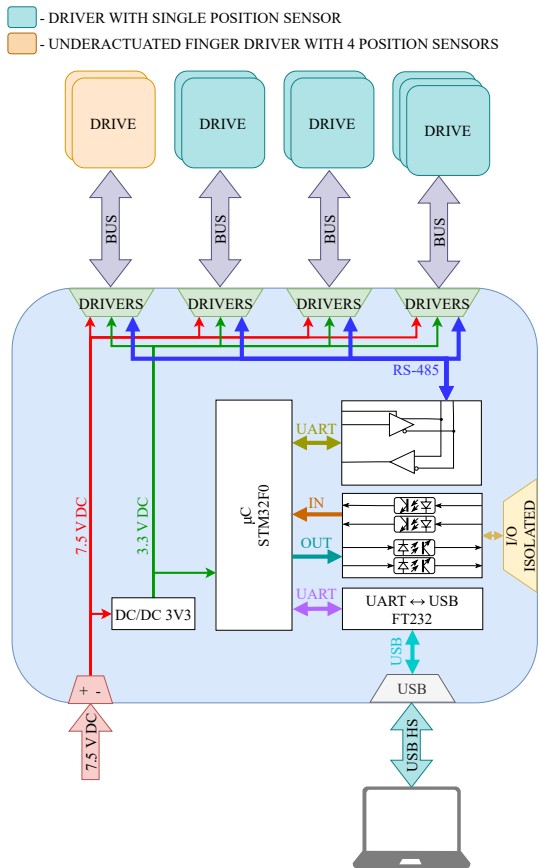

**Figure 9.** PUT-Hand low-level controller architecture, with DRIVER modules connected in star pattern allowing easy interchangeability of components; main unit (HUB) diagram showing circuit components.

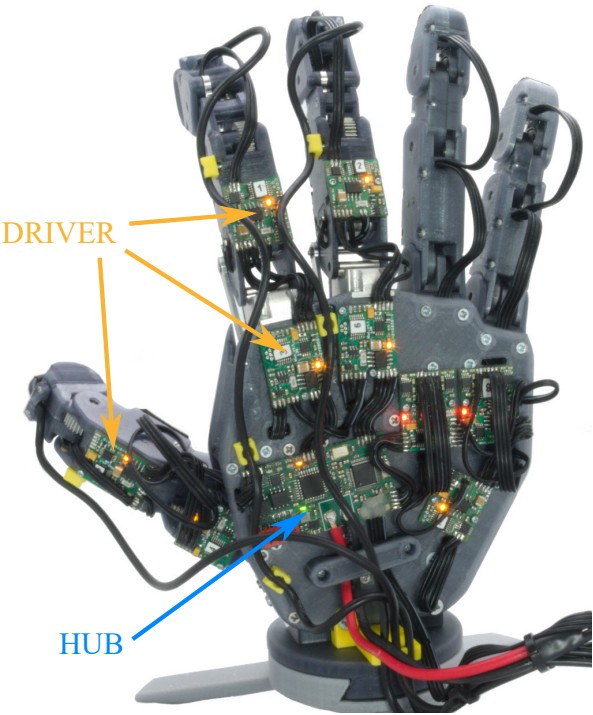

**Figure 10.** PUT-Hand with visible controller electronics; not all DRIVER modules are marked.

### 3.2.1. HUB Design

HUB is an integration unit serving as a bridge between each separate DRIVER and high-level controller. Detailed architecture of the HUB is presented in Figure 9. The unit is controlled by STM32F0 family microcontroller with ARM Cortex-M0 core running at 48 MHz. Communication with PC is carried out using a full-duplex universal asynchronous serial bus (UART), for the ease of use, a UART ⟺ USB converter by FTDI was embedded. The system is powered using 7.5 V DC, HUB includes a DC/DC step-down converter to 3.3 V DC. Both supply voltages are distributed to populate DRIVER connectors. Communication with DRIVER modules is implemented using a half-duplex RS-485 hardware protocol, with a switchable transceiver. Moreover, HUB unit implements two pairs of isolated input and output for additional high-level control purposes, for operating modes without a PC.

### 3.2.2. DRIVER Design

Each DRIVER is a separate servomechanism unit consisting a printed circuit board, a DC motor and resistive position sensors, presented in Figure 11. Two types, with the different number of position sensor connectors, are used in the system. Standard DRIVER supports only one potentiometer, in case of underactuated finger configuration a board with support for four sensors is used. The board features a RS-485 transceiver in half-duplex mode with the a switchable driver. A Texas Instruments DRV8872 MOSFET-based H-bridge with peak 3.6 A current capacity serves as an execution circuit in controlling brushed DC motor. An integrated H-bridge with over-current, under-voltage, and over-temperature protection, including fault interrupt pin is used. High-current side of the motor driver is supplied by 7.5 V DC. Remaining parts of the DRIVER board are supplied by 3.3 V DC. DRIVER also implements a motor current measurement circuit based on a sense resistor. Acquisition of all sensor data, PWM generation, and communication with the HUB unit are carried out by low-power STM32L0 family microcontroller with ARM Cortex-M0+ core running at 32 MHz.

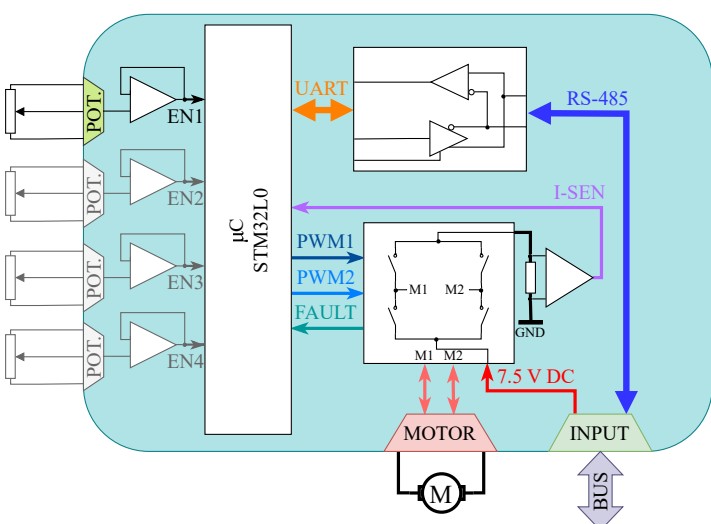

**Figure 11.** Diagram of DRIVER servo unit; two configurations, with one or four resistive position sensors, are used in the system.

### 3.3. Firmware

DRIVER is only an execution unit, and collects sensor data about itself (position, motor current), it does not perform any position control tasks of the drive. Using a defined protocol, secured with a CRC-8 checksum, HUB unit reads the current status of each particular DRIVER and sets the PWM duty for each DC motor, together with the rotation direction. DRIVER modules are addressed using a unique address stored in μC EEPROM memory. In case of a communication loss of over 50 ms,

the DRIVER will engage an electronic motor brake. The protocol uses normalised values for all communication, and all individual drive data is stored internally. This way, a DRIVER can be easily replaced in case of a failure, even with a drive of different type, while remaining transparent to the HUB and overall control scheme.

HUB serves as drive control unit, performing cyclic communication with all DRIVERs at the frequency of 100 Hz. It allows for control of drives in 3 primary modes:

- *Idle*—where all DRIVERs engage electronic brake or disable the H-bridge, depending on user's choice.
- *Internal*—where HUB's internal PID controller with dead-zone is used to position fingers. In this mode, user sets a desired fingers position via the USB interface. Internal PID does not provide force regulation, motor currents are neglected. A diagram of internal control mode is presented in Figure 12a.
- *External*—in this mode HUB acts as a middleman between external user implemented controller and particular drives, providing information about DRIVERs status and forwarding PWM duty. A diagram of external control mode is presented in Figure 12b.

Most communication with PC (high-level controller) is performed on PC request; however, a cyclic status report can be enabled. In this case, the HUB will transmit a full data vector describing hand status (positions, motor currents, mode, etc.) with a frequency of 100 Hz.

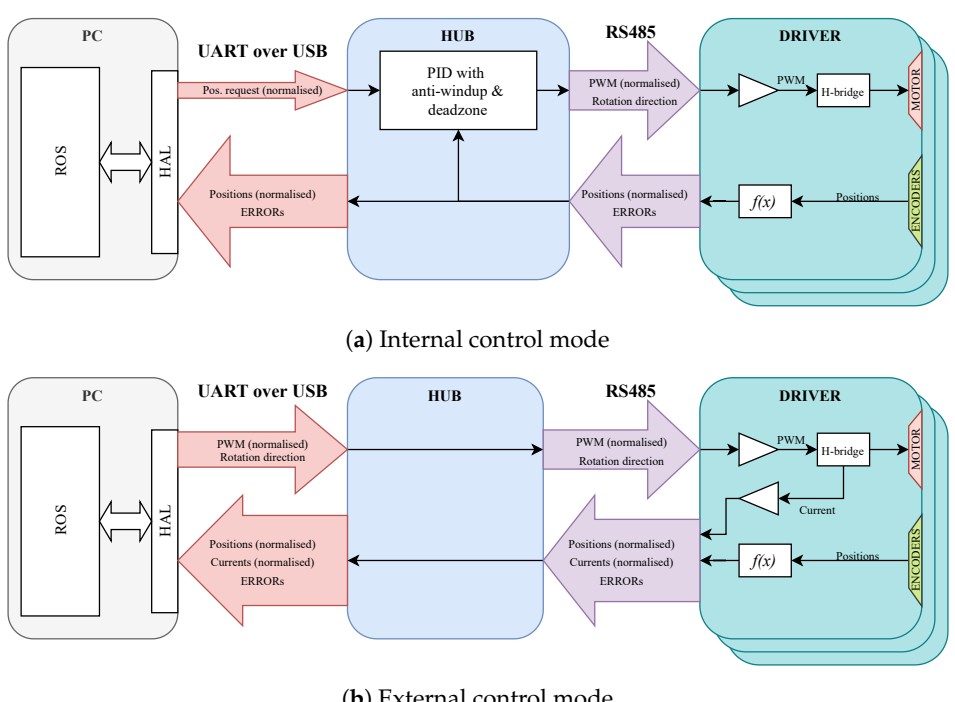

(**a**) Internal control mode

(**b**) External control mode

**Figure 12.** Schematics of PUT-Hand control system in selected modes.

All DRIVER modules feature a calibration procedure, which can be triggered by a high-level controller. During the procedure, the DRIVE module acts independently and moves the joint to both extreme positions to determine position sensor border values and the direction of the motor rotation. Calibration data is then stored in the EEPROM memory of the DRIVE module.

### 3.4. Kinematic Model of PUT-Hand

The kinematic model of the hand is a simplified version of human hand kinematics and a direct result of developed mechanical design. All digits consist of three rotational joints. In the fingers,

the joints (MCP, PIP, and DIP) share a common plane. The fingers' planes are slightly spread outwards to facilitate spherical grasping.

The thumb creates a more complex kinematic chain, with its CMC joint placed at an angle, and the remaining joints (MCP and IP) sharing a common plane. CMC axis orientation was chosen to allow a widest possible range of grasp types and thumb opposition with a simple rotary joint.

Kinematic structure of the gripper together with joint angle naming is presented in Figure 13. All joint ranges are given in Table 4. The range of movements allows the fingers to fully extend and flex (touching palm with the fingertips). CMC range enables full thumb opposition, which makes precision grasping easier (thumb, index, and middle fingers meet each other).

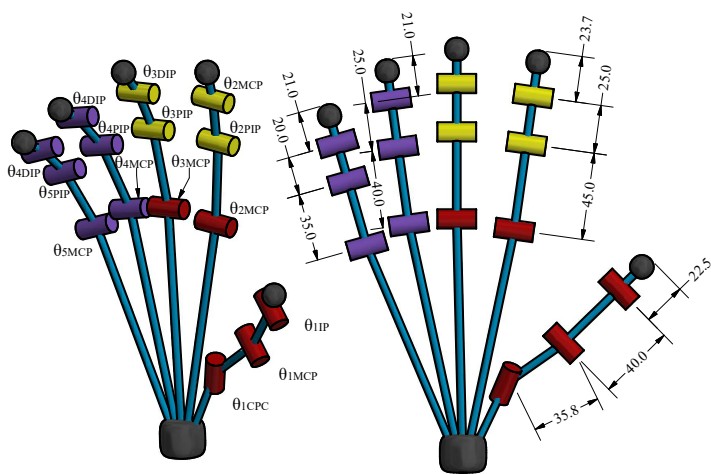

**Figure 13.** PUT-Hand kinematics with basic dimensions and joint naming: red—fully actuated joints, yellow—dependent joints, violet—underactuated joints.

**Table 4.** Joint angle ranges.

| Joint | Max | Min |
|---|---|---|
| $\theta_{1CMC}$ | $0°$ | $135°$ |
| $\theta_{1MCP}$ | $-53.8°$ | $53.8°$ |
| $\theta_{1IP}$ | $0°$ | $90°$ |
| $\theta_{2MCP}/\theta_{3MCP}$ | $7°$ | $90°$ |
| $\theta_{2PIP}/\theta_{3PIP}$ | $0°$ | $96°$ |
| $\theta_{2DIP}/\theta_{3DIP}$ | $0°$ | $74.2°$ |
| $\theta_{4MCP}/\theta_{5MCP}$ | $0°$ | $90°$ |
| $\theta_{4PIP}/\theta_{5PIP}$ | $0°$ | $98.9°$ |
| $\theta_{4DIP}/\theta_{5DIP}$ | $0°$ | $99.8°$ |

### 3.5. High-Level Controller

High-level control of the PUT-Hand is based on the ROS, with core driver written in C++. The driver communicates with the on-board main controller and receives information about the state of the hand and re-sends motion orders. Joint states are cyclically published in the ROS topic, so they can be accessed by multiple programs (ROS nodes) at the same time. In the experiments presented in the article, the arm and the robotic hand are controlled using the ROS environment. The motion of the whole setup is planned using the MoveIt module [69]. The MoveIt module is used to compute forward and inverse kinematic models of the hand, plan the trajectory of the fingers taking into account self-collisions, and execute the trajectory. In the simulated environment and on the real hand we use the position-based interface with the linear joint trajectory that guarantees continuity

at the position level only. In the Gazebo simulation, the hand is controlled by the built-in ROS joint trajectory controller. On the real hand, the ROS driver sends the goal joint positions only. The joint trajectories result from the PID controllers in the DRIVER.

A URDF model of the PUT-Hand was also defined, containing information about kinematic model, visual shape of the links and collision model of the gripper. The URDF model is used in Gazebo simulations. The gripper can be attached to a robotic arm (in our case Universal Robots UR3). Current configuration of the PUT-Hand can be visualised using RViz. The example configuration of the hand and the corresponding visualisation in RViz are presented in Figure 14a,b, respectively.

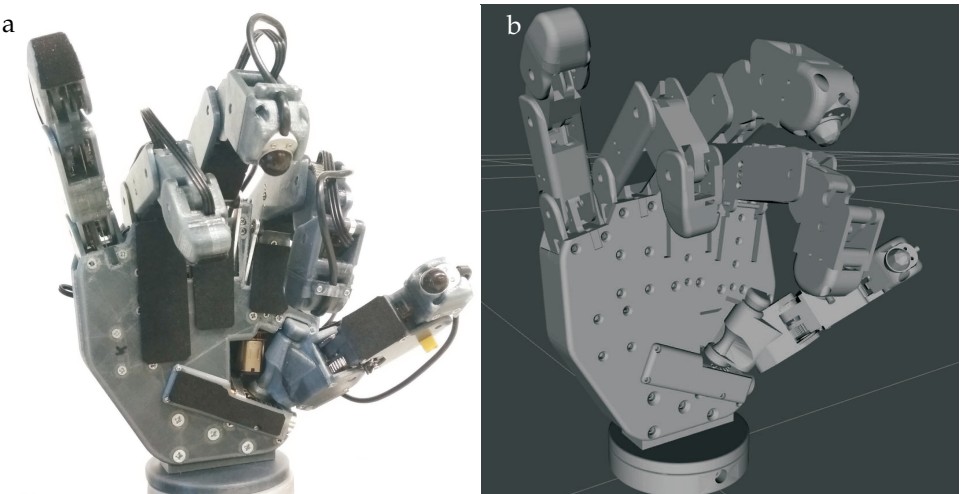

**Figure 14.** Sample configuration of the PUT-Hand (**a**) visualised in the RViz (**b**).

Concurrently, we have defined the most common configurations of the hand, such as initial configuration, open hand, power grasp or pinch grasp. These configurations can be quickly obtained by using predefined commands. We use them to initialise or to show motion capabilities of PUT-Hand or to recover after the error state.

## 4. Results

In this paragraph we first show general capabilities of the hand by performing grasp of different objects. Next, we show the use of the hand for elastic object manipulation in open-loop. Subsequently, we will present the use of force sensors attached to the hand fingertips. After, successful test of manipulation and the use of sensors we performed in-hand manipulation for elastic object identification. Finally, we demonstrate the task of inserting the plug which is often encountered in industrial setting when the cables manipulation is performed.

### 4.1. Grasping

To show kinematic capabilities of the hand, we presented configurations of the hand during grasping of various objects. Example grasps are shown in Figure 15. Objects with various shapes and dimensions were chosen for demonstration: pen, screwdriver, tape, saucer, cup, bottle, plastic plate, ball. We tested precise grasps (Figure 15a,c–f,h) and power grasps (Figure 15b,g,i,j). The hybrid mechanical design of PUT-Hand enables both precision manipulation with fully actuated fingers and stable power grasps with the help of underactuated digits. The underactuated digits have two main advantages. Firstly, they stabilise the position of the large or heavy objects (Figure 15d,g). Secondly, they adapt automatically to the shape of objects without the use of sophisticated tactile feedback and control systems (Figure 15j). The precision grasps with the fully actuated digits are additionally supported by the feedback from tactile sensors mounted on the fingertips of fully actuated fingers.

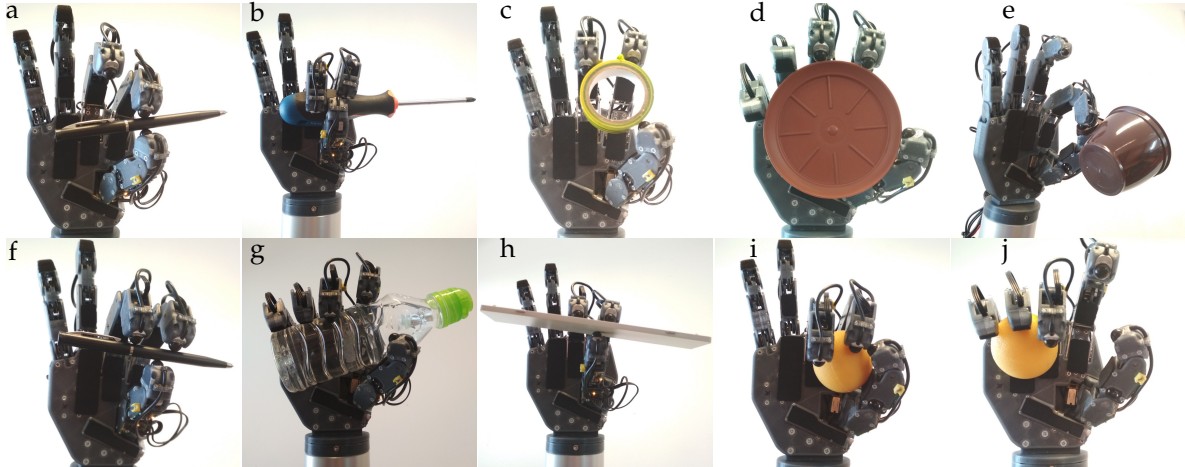

**Figure 15.** Sample configuration of PUT-Hand during grasping of various objects. Examples of precise grasps (**a,c–f,h**), and power grasps (**b,g,i,j**).

## 4.2. Elastic Object Insertion

In the first experiment, we use PUT-Hand attached to a robotic arm (UR3) to bend and force the elastic pipe into a S-shaped channel. The channel has a circular cross section, with an opening at the top narrower than its diameter. Thus, to insert the pipe into the channel the robot has to apply force until the he pipe locks in the channel. The initial configuration of the hand and the pipe are presented in Figure 16a. During the experiment, the robot uses two fully actuated fingers only (index and middle finger).

At the beginning of the experiment, the two fingers are used to force the centre of the pipe into the channel (Figure 16b). Then the procedure uses the index fingertip to bend the pipe (Figure 16c,d). The same procedure is used to bend the second side (Figure 16f–i). Finally, we use a few effective steps to bend the pipe and put it in the channel. The results are presented in Figure 16l. During the experiment, haptic feedback was not used.

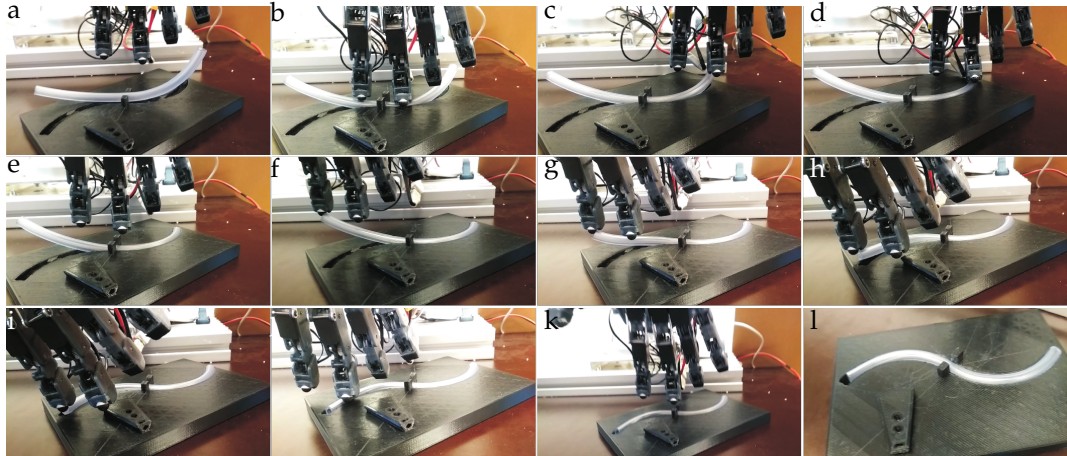

**Figure 16.** Inserting an elastic pipe into a channel using PUT-Hand attached to a UR3 robot: bending and pushing pipe into the channel (**a–k**), final result (**l**).

## 4.3. Contact Force Measurements

In the third set of experiments, we verify usability of the force sensors attached to the fingertips. As the model of the hand is defined in ROS, the direction of contact forces can be easily determined in 3D space. The position of each fingertip and corresponding contact forces can be visualised and are available at any time for the control modules. The example visualisation is available in Figure 17.

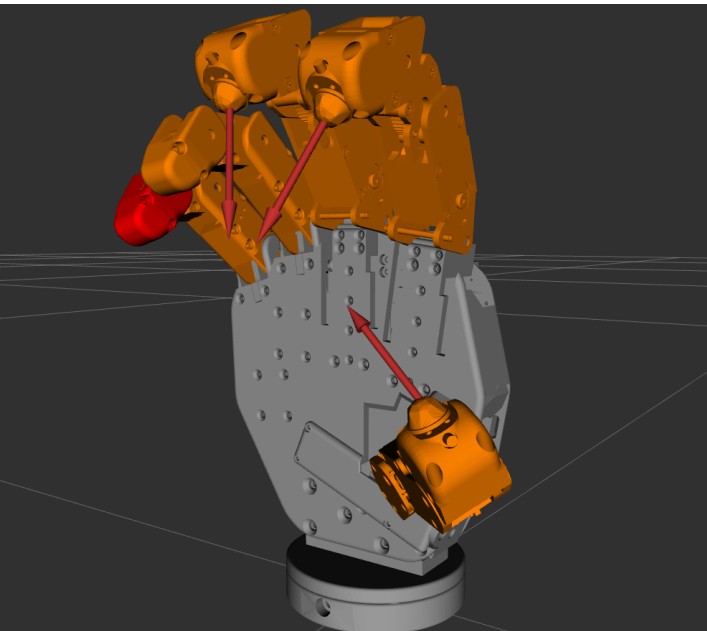

**Figure 17.** RViz visualisation of measured contact forces; red arrow represents the force, with arrow length corresponding to its magnitude

### 4.3.1. In-Hand Elastic Object Identification

In the first contact force experiment, we evaluate the possibility of stable contact event detection using reaction forces. A pinch grasp was performed using fully actuated fingers. The fingers were flexed until a specified force threshold was exceeded. Six objects with various physical properties were used: metal pipe, rubber pipe, squash ball, rubber eraser, and two types of foam. The procedure is presented in Figure 18.

An example trajectory of the thumb tip and corresponding tactile force during pinching of the metal pipe and green foam are presented in Figure 19a and Figure 19b, respectively. Tactile information from the thumb was used because its contact with the object was stable in all of the experiments, and the event of touching the object can be easily distinguished. In case of a rigid object (metal blue pipe), the contact force oscillates after the first contact (Figure 19a). In the experiment with green foam, the soft object dampens the reaction forces and they increase gradually. Obtained force trajectory show a clear difference between a rigid and a soft object and can be used for identification purposes. In both cases, the reaction force stabilises at a constant level (0.8 N for the blue pipe and 0.55 N for the green foam).

We also checked if the force sensors in the hand can identify the properties of manipulated object, such as object stiffness, during in-hand manipulation. After pinching the object, the hand changes its configuration to increase and decrease the contact forces periodically, and the results are logged. A sample relation between contact force and the displacement of the thumb tip during manipulation of the white rubber eraser is presented in Figure 20. The data shows that the displacement was lower than one millimetre, which is too small to determine the object stiffness accurately.

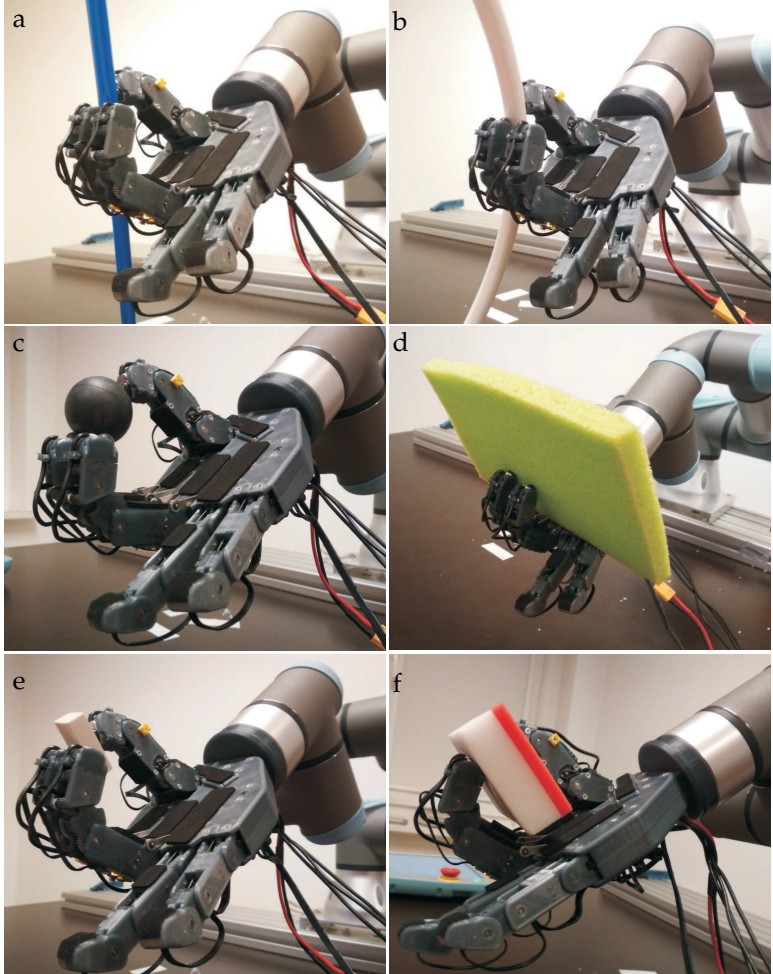

**Figure 18.** Objects grasped by the hand using force feedback: metal pipe (**a**), rubber pipe (**b**), squash ball (**c**), rubber eraser (**e**), and two types of foam (**d,f**).

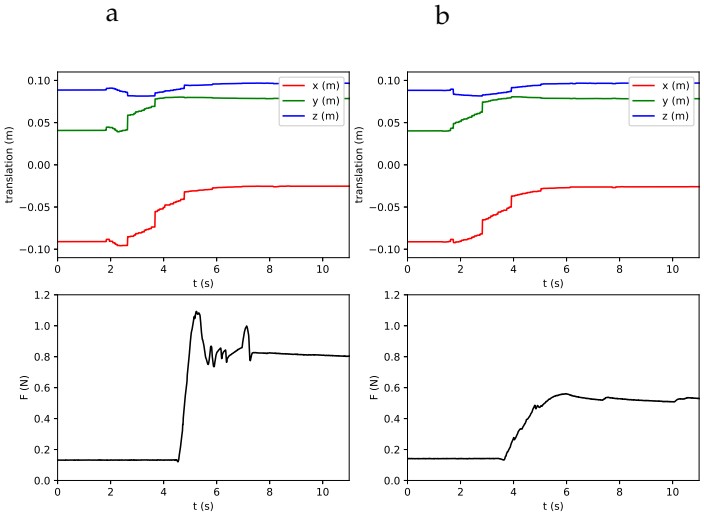

**Figure 19.** Trajectory of the thumb tip and corresponding tactile force trajectory during pinching the blue pipe from Figure 18a (**a**) and green foam from Figure 18d (**b**).

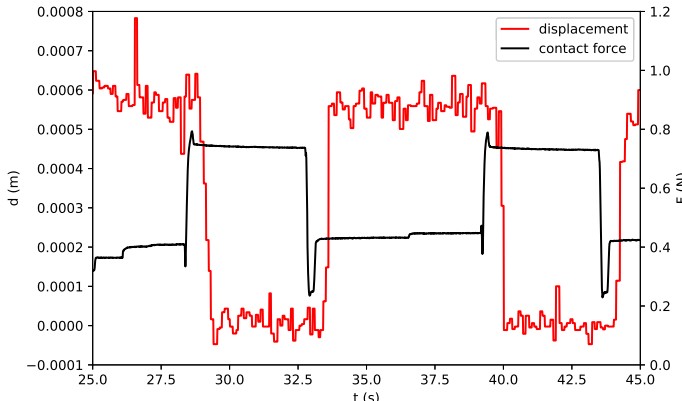

**Figure 20.** Contact force for the thumb and the displacement *d* of the thumb tip during manipulating the white rubber (Figure 18e).

### 4.3.2. Plug Insertion

In the second experiment, a setup consisting of PUT-Hand and UR3 arm performs a task of plug insertion. This simulates a procedure commonly used in automotive industry, widely performed by human operators. Figure 21b shows a plug and socket contraption used during the experiment. Plug contains an elastic component which bends during insertion. Socket is equipped with triangular components which centre the plug and lock it inside.

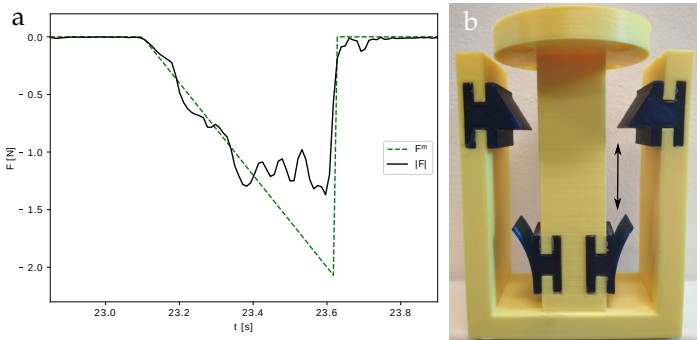

**Figure 21.** Forces (*F*) measured during plug insertion experiment and its predicate values ($F^m$) (**a**); photo of plug and socket pair used during the experiment (**b**).

Figure 22 shows a plug insertion procedure, in which the plug is hold with a tree-point spherical grasp using all tree fully actuated fingers. However, the thumb and the index finger generate opposite forces which hold the plug. The middle finger support the grasp only and prevents the unwanted rotation of the plug. In this experiment, during the insertion itself, main reaction forces are lateral to the tactile sensor base. While forcing two connection elements together, the elastic component of the plug bends, increasing the force required to further move the gripper.

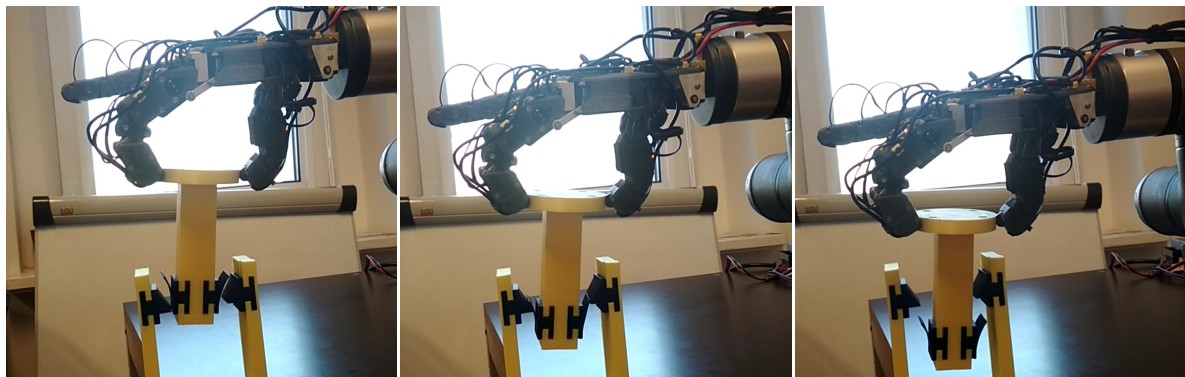

**Figure 22.** UR3 robot equipped with PUT-Hand inserting the plug into the socket.

To model the reaction forces in described setup we use a mechanical system presented in Figure 23. Vertical motion of the plug compresses the spring, increasing a reaction force in the direction opposite to the movement. The reaction force is proportional to the spring constant $k$, friction coefficient $\mu$ and the spring displacement $x$. The compression of the spring $x$ depends on the motion of the plug $y$ and the ramp angle $\theta$:

$$\mathbf{F}^m = 2 \cdot k \cdot \mu \cdot x = 2 \cdot \frac{k \cdot \mu \cdot y}{\tan \theta}, \tag{1}$$

The measured $\theta$ angle and spring constant $k$ is equal to $31°$ and $800 \left[\frac{N}{m}\right]$, respectively. Then, we estimated the friction coefficient $\mu$ to fit the model to the data obtained during the experiment.

Figure 21a presents reaction force $\mathbf{F}^m$ predicted using a model in Figure 23 and those measured during the experiment of plug insertion. The module of the reaction force $|\mathbf{F}|$ in Figure 21a is obtained from the forces measured at the thumb tip $\mathbf{F}^t$ and the index finger tip $\mathbf{F}^i$:

$$|\mathbf{F}| = |\mathbf{F}^t + \mathbf{F}^i|, \tag{2}$$

In the experiment we show that reaction forces which are lateral to the sensor can be measured using the given configuration of the sensors in the fingertips. The middle finger is used only to stabilise the grasp and the tactile sensor does not touch the object during the experiment. We model the static properties of the system only thus the force decreases instantly when the plug is in the socket. In practice the mass of the system and elasticities cause the gradual decrease of the reaction forces shown in Figure 21a.

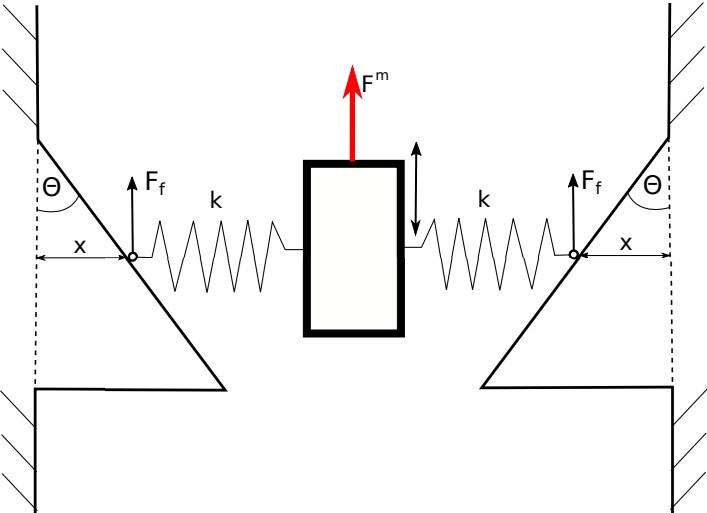

**Figure 23.** Mechanical model of plug insertion experimental setup.

## 5. Conclusions and Further Development

In this article, we propose a design of a five-finger anthropomorphic gripper intended for elastic object manipulation. All design files are published at https://github.com/puthand. Main contributions of this work are:

- open-source mechanical design of PUT-Hand, a hybrid anthropomorphic gripper design consisting of three fully actuated fingers (thumb, index, and middle) for precise manipulation, and two underactuated tendon-driven digits (ring and small) for power-grasp support;
- open-source on-board controller design and firmware;
- sensory system using optical 3-axis force sensors;
- ROS-based open-source driver for high-level control including motion planning and visualisation;
- experimental verification presenting mechanical and sensory capabilities of the proposed design.

PUT-Hand is a hybrid design, taking advantages of both fully actuated [1,5,11] and underactuated [6,24] designs. Fully actuated fingers can be used for precise grasping (Figure 15) and interaction with elastic objects (Figure 16), while underactuated fingers stabilise the grasp during dealing with heavy objects such as bottles (Figure 15g) or large objects.

The gripper is equipped with three tri-axial force sensors mounted on fingertips of fully actuated fingers, allowing for measurement of magnitude and direction of contact forces. This information can be used to measure the physical properties of the objects the robot is interacting with. Moreover, with simple modelling, we can detect the state of the environment (Figure 22).

Future work includes integration of presented setup (robot equipped with PUT-Hand gripper) with a visual perception system. Combination of visual and tactile feedback can be used to further increase the autonomy of the robot, and improve the performance of grasping, modelling and manipulation of elastic objects.

**Author Contributions:** Conceptualisation, T.M., J.T. and D.B.; Funding acquisition, K.W.; Investigation, T.M., J.T. and D.B.; Methodology, D.B.; Project administration, K.W.; Resources, K.W.; Software, T.M., J.T. and D.B.; Writing—original draft, T.M., J.T. and D.B.; Writing—review & editing, K.W. All authors have read and agreed to the published version of the manuscript.

**Funding:** This work is supported by grant No. LIDER/3/0183/L-7/15/NCBR/2016 funded by The National Centre for Research and Development (Poland)

**Conflicts of Interest:** The authors declare no conflict of interest.

## Abbreviations

The following abbreviations are used in this manuscript:

| | |
|---|---|
| DIP | Distant Interphalangeal joint |
| MCP | Metacarpophalangeal joint |
| IP | Interphalangeal joint |
| ROS | Robot Operating System |
| DoF | Degree of Freedom |
| URDF | Universal Robot Description Format |

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
