# Peer review of "PUT-Hand—Hybrid Industrial and Biomimetic Gripper for Elastic Object Manipulation"

_electronics, doi:10.3390/electronics9071147_

Round 1

Reviewer 1 Report

This article designed a five-fingered anthropomorphic gripper which is presented specifically designed for elastic objects manipulation including grasping, tactile sensing, and application-oriented tasks. The manipulator features a hybrid design, being equipped with three fully-actuated fingers for precise manipulation, and two underactuated, tendon-driven digits for secure power grasping. The contents are detailed and the hand works well. However, before publication, a minor revise is required to address the following issues.

  1. As this article introduced, the manipulator features a hybrid design being equipped with three fully-actuated fingers for precise manipulation, and two underactuated, tendon-driven digits for secure power grasping. However, in the grasping experiment, I find that there are two grasping methods. One is applying three fully-actuated fingers, and the other is applying one fully-actuated finger and two underactuated. Better to include comparison between these two method.
  2. The manuscript need to compare the presented hand with the typical existing robotic hand.
  3. As we can see in experiment part, most of the work are conducted by three fully-actuated fingers. Why the hand is designed to be five-fingered? Looks like a human hand. It’s important, but not scientific enough. More grasping modes should be included.
  4. For manipulating elastic object, how to define the manipulation is successful? Compared with real human grasping, what’s the difference? How to learn from the real human grasping.

Reviewer 2 Report

The paper deals with a novel hybrid industrial and biomimetic gripper for elastic object manipulation. The paper is interesting and well structured. However, the following points should be improved before considering the paper suitable for publication.

1) Since the journal deals with electronics, the electrical and electronic aspects should be better analyzed. What are the limitation of the proposed device from the electronic point of view?

2) The authors should better describe the control system and data acquisition that have been implemented for the proposed device. A block diagram of the control system would be helpful for the reader.

3) It is not clear if both the robot and the robotic hand are controlled in ROS environment. The trajectroy planning approach is not described (velocity and acceleration of the robot during motion).

4) The authors state that some springs are placed at the joints of the fingers. It is not clear how the stiffness and the location of these springs have been chosen. Are these springs useful to reduce the energy consumprion of the robotic hand? (Please see: Scalera, L., Palomba, I., Wehrle, E., Gasparetto, A., & Vidoni, R. (2019). Natural motion for energy saving in robotic and mechatronic systems. Applied Sciences9(17), 3516.)

Round 2

Reviewer 2 Report

The paper was improved by following my comments and suggestions.